# Geometric phase magnetometry using a solid-state spin

K. Arai[1], J. Lee[1], C. Belthangady[2,3], D.R. Glenn[2,3,4], H. Zhang[2,3] & R.L. Walsworth[2,3,4]

A key challenge of magnetometry lies in the simultaneous optimization of magnetic field sensitivity and maximum field range. In interferometry-based magnetometry, a quantum two-level system acquires a dynamic phase in response to an applied magnetic field. However, due to the $2\pi$ periodicity of the phase, increasing the coherent interrogation time to improve sensitivity reduces field range. Here we introduce a route towards both large magnetic field range and high sensitivity via measurements of the geometric phase acquired by a quantum two-level system. We experimentally demonstrate geometric-phase magnetometry using the electronic spin associated with the nitrogen vacancy (NV) color center in diamond. Our approach enables unwrapping of the $2\pi$ phase ambiguity, enhancing field range by 400 times. We also find additional sensitivity improvement in the nonadiabatic regime, and study how geometric-phase decoherence depends on adiabaticity. Our results show that the geometric phase can be a versatile tool for quantum sensing applications.

[1] Department of Physics, Massachusetts Institute of Technology, Cambridge, MA 02139, USA. [2] Harvard-Smithsonian Center for Astrophysics, Cambridge, MA 02138, USA. [3] Department of Physics, Harvard University, Cambridge, MA 02138, USA. [4] Center for Brain Science, Harvard University, Cambridge, MA 02138, USA. These authors contributed equally K. Arai, J. Lee. Correspondence and requests for materials should be addressed to R.L.W. (email: rwalsworth@cfa.harvard.edu)

The geometric phase[1,2] plays a fundamental role in a broad range of physical phenomena[3–5]. Although it has been observed in many quantum platforms[6–9] and is known to be robust against certain types of noise[10,11], geometric phase applications are somewhat limited, including certain protocols for quantum simulation[12,13] and computation[14–17]. However, when applied to quantum sensing, e.g., of magnetic fields, unique aspects of the geometric phase can be exploited to allow realization of both good magnetic field sensitivity and large field range in one measurement protocol. This capability is in contrast to conventional dynamic-phase magnetometry, where there is a trade-off between sensitivity and field range. In dynamic-phase magnetometry using a two-level system (e.g., two spin states), the amplitude of an unknown magnetic field $B$ can be estimated by determining the relative shift between two energy levels induced by that field (Methods). A commonly used approach is to measure the dynamic phase accumulated in a Ramsey interferometry protocol. An initial resonant $\pi/2$ pulse prepares the system in a superposition of the two levels. In the presence of an external static magnetic field $B$ along the quantization axis, the system evolves under the Hamiltonian $H = \hbar \gamma B \sigma_z / 2$, where $\gamma$ denotes the gyromagnetic ratio and $\sigma_z$ is the $z$-component of the Pauli spin vector. During the interaction time $T$ (limited by the spin dephasing time $T_2^*$), the Bloch vector $\mathbf{s}(t)$ depicted on the Bloch sphere precesses around the fixed Larmor vector $\mathbf{R} = (0, 0, \gamma B)$, and acquires a dynamic phase $\phi_d = \gamma BT$. The next $\pi/2$ pulse maps this phase onto a population difference $P = \cos \phi_d$, which can be measured to determine $\phi_d$ and hence the magnetic field $B$ (Supplementary Note 1).

Such dynamic-phase magnetometry possesses two well-known shortcomings. First, the sinusoidal variation of the population difference with magnetic field leads to a $2\pi$ phase ambiguity in interpretation of the measurement signal and hence determination of $B$. Specifically, since the dynamic phase is linearly proportional to the magnetic field, for any measured signal $P_{meas}$ (throughout the text, this value corresponds to $(\Delta FL/FL) \times k$, where $k$ is a constant that depends on NV readout contrast), there are infinite magnetic field ambiguities: $B_m = (\gamma T)^{-1} (\cos^{-1} P_{meas} + 2\pi m)$, where $m = 0, \pm 1, \pm 2 \ldots \pm \infty$. Thus, the range of magnetic field amplitudes that one can determine without modulo $2\pi$ phase ambiguity is limited to one cycle of oscillation: $B_{max} \propto 1/T$ (Supplementary Note 2, Supplementary Figure 5). Second, there is a trade-off between magnetic field sensitivity and field range, as the interaction time also restricts the shot-noise-limited magnetic field sensitivity: $\eta \propto 1/T^{1/2}$. Consequently, an improvement in field range via shorter $T$ comes at the cost of a degradation in sensitivity (Supplementary Note 3). Use of a closed-loop lock-in type measurement[18], quantum phase estimation algorithm[19,20], or non-classical states[21,22] can alleviate these disadvantages; however, such approaches require either a continuous measurement scheme with limited sensitivity, large resource overhead (additional experimental time) or realization of long-lived entangled or squeezed states.

In the present work, we use the electronic spin associated with a single nitrogen vacancy (NV) color center in diamond to demonstrate key advantages of geometric-phase magnetometry: (i) it resolves the $2\pi$ phase ambiguity limiting dynamic-phase magnetometry; and (ii) it decouples magnetic field range and sensitivity, leading to a 400-fold enhancement in field range at constant sensitivity in our experiment. We also show additional improvement of magnetic field sensitivity in the nonadiabatic regime of mixed geometric and dynamic-phase evolution. By employing a power spectral density analysis[23], we find that adiabaticity plays an important role in controlling the degree of coupling to environmental noise and hence the spin coherence timescale.

## Results

**Geometric-phase magnetometry protocol.** To implement geometric-phase magnetometry, we use a modified version of an experimental protocol ("Berry sequence") previously applied to a superconducting qubit[9]. In our realization, the NV spin sensor is placed in a superposition state by a $\pi/2$ pulse, where the driving frequency of the $\pi/2$ pulse is chosen to be resonant with the NV $m_s = 0 \leftrightarrow m_s = +1$ transition at constant bias field $B_{bias}$ ($\approx$9.6 mT in our experiment) aligned with the NV axis. A small signal field $B$ (~100 $\mu$T in our experiment) is then applied parallel to $B_{bias}$, and the NV spin acquires a geometric phase due to off-resonant microwave driving with control parameters cycled along a closed path as illustrated in Fig. 1b (Methods). Under the rotating wave approximation, the effective two-level Hamiltonian is given by:

$$H = \frac{\hbar}{2} \left( \Omega \cos(\rho)\sigma_x + \Omega \sin(\rho)\sigma_y + \gamma B \sigma_z \right). \tag{1}$$

Here, $\Omega$ is the NV spin Rabi frequency for the microwave driving field, $\rho$ is the phase of the driving field, and $\boldsymbol{\sigma} = (\sigma_x, \sigma_y, \sigma_z)$ is the Pauli spin vector. By sweeping the phase, the Larmor vector $\mathbf{R}(t) = R^\star(\sin\theta \cos\rho, \sin\theta \cos\rho, \cos\theta)$, where $\cos\theta = \gamma B/(\Omega^2 + (\gamma B)^2)^{1/2}$, $R = (\Omega^2 + (\gamma B)^2)^{1/2}$, rotates around the $z$-axis. The Bloch vector $\mathbf{s}(t)$ then undergoes precession around this rotating Larmor vector (for detailed picture of the measurement protocol, see Supplementary Fig 2). If the rotation is adiabatic (i.e., adiabaticity parameter $A \equiv \dot{\rho} \sin \theta / 2R \ll 1$), then the system acquires a geometric phase proportional to the product of (i) the solid angle $\Theta = 2\pi(1 - \cos\theta)$ subtended by the Bloch vector trajectory and (ii) the number of complete rotations $N$ of the Bloch vector around the Larmor vector in the rotating frame defined by the frequency of the initial $\pi/2$ pulse. We apply this Bloch vector rotation twice during the interaction time $T$, with alternating direction separated by a $\pi$ pulse, which cancels the accumulated dynamic phase and doubles the geometric phase: $\phi_g = 2N\Theta$ (Supplementary Note 1). A final $\pi/2$ pulse allows this geometric phase to be determined from standard fluorescence readout of the NV spin-state population difference:

$$P_{meas}(B) = \cos\left[ 4\pi N \left( 1 - \frac{\gamma B}{\sqrt{(\gamma B)^2 + \Omega^2}} \right) \right]. \tag{2}$$

This normalized geometric-phase signal (Supplementary Note 1) exhibits chirped oscillation as a function of magnetic field. There are typically only a small number of field ambiguities that give the same signal $P_{meas}$; these can be resolved uniquely by measuring the slope $dP_{meas}/dB$ (Supplementary Note 2, Supplementary Fig. 5). From the form of Eq. (2) it is evident that at large $B$, cosine signal approaches to zero like $B^{-2}$, and the slope goes to zero. Hence, we define the field range as the largest magnetic field value ($B_{max}$) that gives the last oscillation minimum in the signal: $B_{max} \propto \Omega\ N^{1/2}$. Importantly, the field range of geometric-phase magnetometry has no dependence on the interaction time $T$. If the magnetic field is below $B_{max}$, then one can make a geometric-phase magnetometry measurement with optimal sensitivity $\eta \propto \Omega\ N^{-1}\ T^{1/2}$ (Supplementary Note 3).

**Comparison between dynamic- and geometric-phase magnetometry.** We implemented both dynamic- and geometric-phase magnetometry using the optically addressable electronic spin of a single NV color center in diamond (Fig. 2a) (Supplementary Figs. 1-3). NV-diamond magnetometers provide high spatial

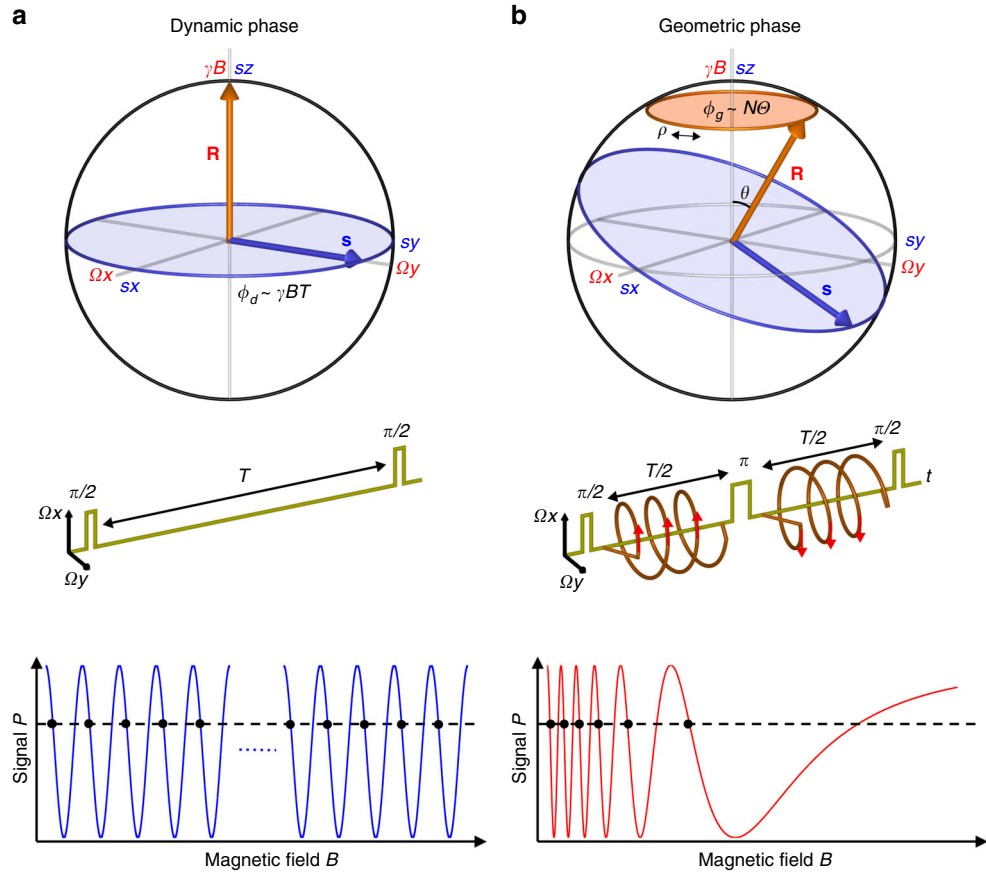

**Fig. 1** Concepts of dynamic- and geometric-phase magnetometry. **a** For dynamic-phase magnetometry with an NV spin, the Bloch vector $\mathbf{s} = (s_x, s_y, s_z)$ (blue arrow), initially prepared by a $\pi/2$ pulse in a superposition state between two levels, precesses about the fixed Larmor vector $\mathbf{R} = (0, 0, \gamma B)$ (red arrow). During the interaction time $T$ between the two $\pi/2$ pulses, the spin coherence accumulates a dynamic phase $\phi_d = \gamma BT$, equivalent to the angle swept by the Bloch vector on the equator. The phase is then mapped by a second $\pi/2$ pulse to a population difference signal $P = \cos\phi_d$, which is measured optically. Due to a $2\pi$ phase periodicity, an infinite number of magnetic field values (black dots) give the same signal, leading to an ambiguity. **b** For geometric-phase magnetometry with an NV spin, a Berry sequence is employed. The Bloch vector is first prepared by a $\pi/2$ pulse in a superposition state between two levels. An additional off-resonant driving is then used to rotate the Larmor vector about the z-axis $N$ times, $\mathbf{R}(t) = (\Omega\cos\rho(t), \Omega\sin\rho(t), \gamma B)$, where $\rho(t) = 4\pi Nt/T$. The spin coherence acquires a geometric phase $\phi_g = N\Theta$, proportional to the number of rotations $N$ and the solid angle $\Theta = 2\pi(1 - \cos\theta)$ subtended by the trajectory of the Larmor vector. To cancel the dynamic phase and double the geometric phase, the direction of rotation is alternated before and after a $\pi$ pulse at the midpoint of the interaction time. At the end of the Berry sequence, the phase is mapped by a second $\pi/2$ pulse to a population difference signal $P = \cos\phi_g$, which is measured optically. The signal exhibits chirped oscillation with magnetic field amplitude, which yields at most finite magnetic field degeneracies (black dots). The signal vs. field slope resolves this ambiguity

resolution under ambient conditions[24–26], and have therefore found wide-ranging applications, including in condensed matter physics[27,28], the life sciences[29,30], and geoscience[31]. At an applied bias magnetic field of 9.6 mT, the degeneracy of the NV $m_s = \pm 1$ levels is lifted. The two-level system used in this work consists of the ground state magnetic sublevels $m_s = 0$ and $m_s = +1$, which can be coherently addressed by applied microwave fields. The hyperfine interaction between the NV electronic spin and the host $^{14}$N nuclear spin further splits the levels into three states, each separated by 2.16 MHz. Upon green laser illumination, the NV center exhibits spin-state-dependent fluorescence and optical pumping into $m_s = 0$ after a few microseconds. Thus, one can prepare the spin states and determine the population by measuring the relative fluorescence (see Methods for more details).

First, we performed dynamic-phase magnetometry using a Ramsey sequence to illustrate the $2\pi$ phase ambiguity and show how the dependence on interaction time gives rise to a trade-off between field range and magnetic field sensitivity. We recorded the NV fluorescence signal as a function of the interaction time $T$

between the two microwave $\pi/2$ pulses (Fig. 1a). Signal contributions from the three hyperfine transitions of the NV spin result in the observed beating behavior seen in Fig. 2b. We fixed the interaction time at $T = 0.2, 0.5, 1.0$ μs, varied the external magnetic field for each value of $T$, and observed a periodic fluorescence signal with a $2\pi$ phase ambiguity (Fig. 2c). The oscillation period decreased as the interaction time was increased, indicating a reduction in the magnetic field range (i.e., smaller $B_{max}$). In contrast, the magnetic field sensitivity, which depends on the maximum slope of the signal, improved as the interaction time increased. For each value of $T$, we fit the fluorescence signal to a sinusoid dependent on the applied magnetic field and extracted the oscillation period and slope, which we used to determine the experimental sensitivity and field range. From this procedure, we obtained $\eta \propto T^{-0.49(6)}$ and $B_{max} \propto T^{-0.96(2)}$, consistent with expectations for dynamic-phase magnetometry and illustrative of the trade-off inherent in optimizing both $\eta$ and $B_{max}$ as a function of interaction time (Supplementary Fig. 7).

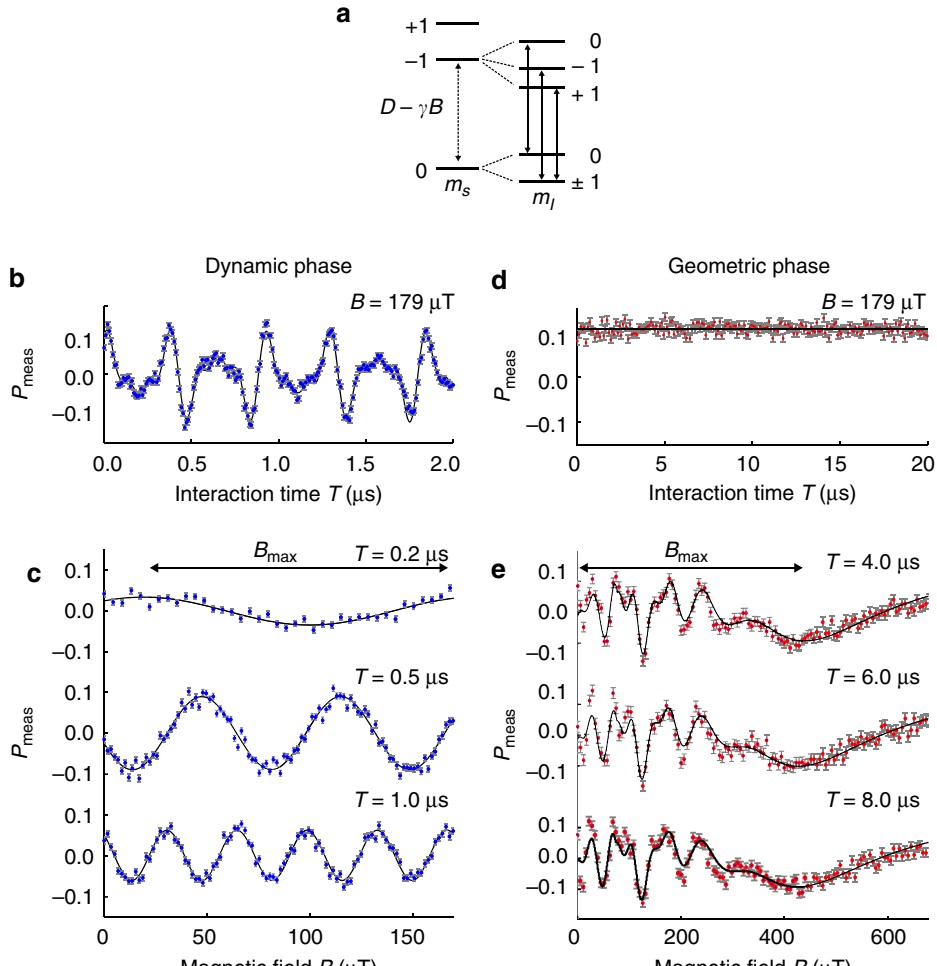

**Fig. 2** Demonstration of dynamic- and geometric-phase magnetometry using a single NV spin in diamond. **a** NV electronic spin ($S = 1$) has sublevels $m_s =$ 0 and $\pm 1$ with zero-field splitting $D = 2\pi \times 2.87$ GHz. An external magnetic field $B$ introduces Zeeman splitting between the $m_s = \pm 1$ states with gyromagnetic ratio $\gamma = 2\pi \times 28$ GHz T$^{-1}$. $m_s = 0$ and $-1$ define the two-level system used in this work. Hyperfine interactions with the host $^{14}$N nuclear spin lead to $m_I = 0, \pm 1$, split by $\pm 2.16$ MHz. **b**–**e** Blue and red dots represent measured magnetometry data for dynamic phase (**b**, **c**) and geometric phase (**d**, **e**) protocols, respectively. Vertical axes give the measured optical signal $P_{\text{meas}} = (\Delta FL/FL) \times k$, where $\Delta FL/FL$ is the fractional change of NV-spin-state-dependent fluorescence and $k$ is a constant that depends on NV readout contrast. Error bars are one standard deviation photon shot-noise. Black lines show fits to a model outlined in the main text. Blue- and red-shaded regions represent maximum magnetic field ranges. Beating due to three hyperfine resonances is evident in **b**. In dynamic-phase magnetometry, the oscillation period decreases as the interaction time increases, indicating a trade-off between sensitivity and field range (**c**). Geometric-phase magnetometry signal in (**d**) shows independence of $T$. Field range is defined at the last minimum (**e**)

Next, we used a Berry sequence to demonstrate two key advantages of geometric-phase magnetometry: i.e., there is neither a $2\pi$ phase ambiguity nor a sensitivity/field-range trade-off with respect to interaction time. For fixed adiabatic control parameters of $\Omega/2\pi = 5$ MHz and $N = 3$, the observed geometric-phase magnetometry signal $P_{\text{meas}}$ has no dependence on interaction time $T$ (Fig. 2d). Varying the external magnetic field with fixed interaction times $T = 4.0, 6.0, 8.0$ µs, $P_{\text{meas}}$ exhibits identical chirped oscillations for all $T$ values (Fig. 2e), as expected from Eq. (2). From the $P_{\text{meas}}$ data we extract d$P_{\text{meas}}$/d$B$, which allows us to determine the magnetic field uniquely for values within the oscillatory range (Supplementary Note 2), and also to quantify $B_{\text{max}}$ from the last minimum point of the chirped oscillation (Fig. 2e). Additional measurements of the dependence of $P_{\text{meas}}$ on the adiabatic control parameters $\Omega$, $N$, and $T$ (Supplementary Figs. 4, 6) yield the scaling of sensitivity and field range: $\eta \propto \Omega^{1.2(5)} N^{-0.92(1)} T^{0.46(1)}$ and $B_{\text{max}} \propto \Omega^{0.9(1)} N^{0.52(5)} T^{0.02}$

(1), which is consistent with expectations and shows that geometric-phase magnetometry allows $\eta$ and $B_{\text{max}}$ to be independently optimized as a function of interaction time (Supplementary Fig. 7).

In Fig. 3 we compare the measured sensitivity and field range for geometric-phase and dynamic-phase magnetometry. For each point displayed, the sensitivity is measured directly at small $B$ ($0.01 \sim 0.1$ mT), whereas the field range is calculated from the measured values of $N$ and $\Omega$ (for geometric-phase magnetometry) and $T$ (for dynamic-phase magnetometry, with $T$ limited by the dephasing time $T_2^*$), following the scaling laws give above. Since geometric-phase magnetometry has three independent control parameters ($T$, $N$, and $\Omega$), $B_{\text{max}}$ can be increased without changing sensitivity by increasing $N$ and $\Omega$ while keeping the ratio $N/\Omega$ fixed. Such "smart control" allows a tenfold improvement in geometric-phase sensitivity (compared to dynamic-phase measurements) for $B_{\text{max}} \sim 1$ mT, and a 400-fold enhancement

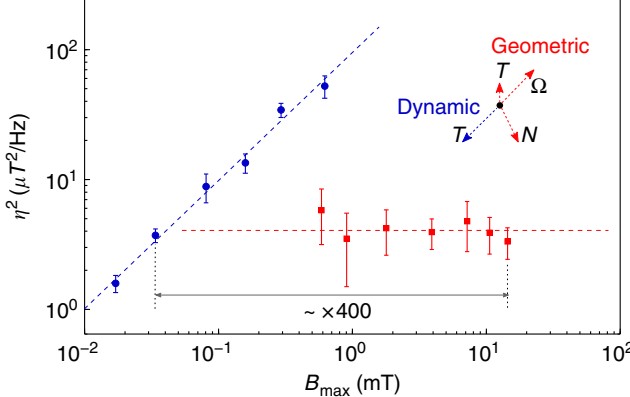

**Fig. 3** Decoupling of magnetic field sensitivity and maximum field range. Measured performance of dynamic-phase (blue dots) and geometric-phase (red squares) magnetometry. Dashed lines are linear fits to data. Dashed arrows indicate the orientation of control parameters $\Omega$, $N$, $T$ as independent vectors on the ($\eta^2$, $B_{max}$) map. Since a Ramsey sequence used for dynamic-phase magnetometry has only a single control parameter ($T$), the relations for sensitivity ($\eta \propto T^{1/2}$) and field range ($B_{max} \propto T^{-1}$) are unavoidably coupled as $\eta^2 \propto B_{max}$. In contrast, a Berry sequence used for geometric-phase magnetometry employs all three control parameters, and thus the sensitivity ($\eta \propto \Omega^{-1} N T^{1/2}$) can be chosen independently of the field range ($B_{max} \propto \Omega N^{1/2} T^0$). For example, larger $B_{max}$ with constant $\eta$ is obtainable with geometric-phase magnetometry by increasing $\Omega$ and $N$ while keeping $T$ and the ratio $\Omega/N$ fixed. Error bars represent one standard deviation of the results

$B_{max}$ at a sensitivity of ~2 μT Hz$^{-1/2}$. Similarly, the sensitivity can be improved without changing $B_{max}$ by decreasing the interaction time, with a limit set by the adiabaticity condition ($A \equiv \dot{\rho} \sin\theta/2R \approx N/\Omega T \ll 1$).

**Geometric-phase magnetometry in nonadiabatic regime.** Finally, we explored geometric-phase magnetometry outside the adiabatic limit by performing Berry sequence experiments and varying the adiabaticity parameter by more than two orders of magnitude (from $A \approx 0.01-5$). We find good agreement between our measurements and simulations, with an onset of nonadiabatic behavior for $A \gtrsim 0.2$ (Supplementary Figure 8). At each value of the adiabaticity parameter $A$, we determine the magnetic field sensitivity from the largest slope of the measured magnetometry curve. (The magnetometry curve is the plot of $P_{meas}$ obtained as a function of applied magnetic field $B$.) To compare with the best sensitivity provided by dynamic-phase magnetometry, we fix the interaction time at $T \approx T_2^\star/2$ in the nonadiabatic geometric-phase measurements. We find that the sensitivity of geometric-phase magnetometry improves in the nonadiabatic regime, and becomes smaller than the sensitivity from dynamic-phase measurements for $A \gtrsim 1.0$ (Fig. 4a).

To understand this behavior, we recast the sensitivity scaling in terms of the adiabaticity parameter and interaction time, $\eta \propto A^{-1} T^{-1/2}$ and investigated the trade-off between these parameters. (Note that in the nonadiabatic regime the Bloch vector no longer strictly follows the Larmor vector, and thus the sensitivity scaling is not exact.) We performed a spectral density analysis to assess how environmental noise leads to both dynamic- and geometric-phase decoherence, with the relative contribution set by the adiabaticity parameter $A$, thereby limiting the interaction time $T$. We take the exponential decay of the NV spin coherence $W(T) \sim \exp(-\chi(T))$, characterized by the decoherence function $\chi(T)$ given by

$$\chi(T) = A^2 \int_0^\infty \frac{d\omega}{\pi} S(\omega) \frac{F_0(\omega T)}{\omega^2} + \int_0^\infty \frac{d\omega}{\pi} S(\omega) \frac{F_1(\omega T)}{\omega^2}. \quad (3)$$

Here, $S(\omega)$ is a spectral density function that describes magnetic noise from the environment; $F_0(\omega T) = 2\sin^2(\omega T/2)$ is the filter function for geometric-phase evolution in the Berry sequence, which is spectrally similar to a Ramsey sequence, with maximum sensitivity to static and low frequency ($\lesssim 1/T$) magnetic fields; and $F_1(\omega T) = 8\sin^4(\omega T/4)$ is the filter function for dynamic-phase evolution in the Berry sequence, which is spectrally similar to a Hahn-echo sequence, with maximum sensitivity to higher frequency ($\gtrsim 1/T$) magnetic fields (Supplementary Note 4).

**Geometric-phase coherence time.** Figure 4b shows examples of the measured decay of the geometric-phase signal ($P_{meas}$) as a function of interaction time $T$ and adiabaticity parameter $A$. From such data we extract the geometric-phase coherence time $T_{2g}$ by fitting $P_{meas} \sim \exp[-(T/T_{2g})^2]$. We observe four regimes of decoherence behavior (Fig. 4c), which can be understood from Eq. (3) and its schematic spectral representation in Fig. 4d. For $A < 0.1$ (adiabatic regime), dynamic-phase evolution (i.e., Hahn-echo-like behavior) dominates the decoherence function $\chi(T)$ and thus $T_{2g} \sim T_2 \approx 500$ μs. For $0.1 \leq A < 1.0$ (intermediate regime), the coherence time is inversely proportional to the adiabaticity parameter ($T_{2g} \sim T_2^\star/A$) as geometric-phase evolution (with Ramsey-like dephasing) becomes increasingly significant. For $A \approx 1.0$ (nonadiabatic regime), geometric-phase evolution dominates $\chi(T)$ at long times and thus $T_{2g} \sim T_2^\star \approx 50$ μs. For $A \gg 1.0$ (strongly nonadiabatic limit), the driven rotation of the Larmor vector is expected to average out during a Berry sequence (Fig. 1b) and only the $z$-component of the Larmor vector remains. Thus, the Berry sequence converges to a Hahn-echo-like sequence and the coherence time is expected to increase to $T_2$ for very large $A$.

**Discussion**

In summary, we demonstrated an approach to NV-diamond magnetometry using geometric-phase measurements, which avoids the trade-off between magnetic field sensitivity and maximum field range that limits traditional dynamic-phase magnetometry. For an example experiment with a single NV, we realize a 400-fold enhancement in static (DC) magnetic field range at constant sensitivity. We also explored geometric-phase magnetometry as a function of adiabaticity, with good agreement between measurements and model simulations. We find that adiabaticity controls the coupling between the NV spin and environmental noise during geometric manipulation, thereby determining the geometric-phase coherence time. Furthermore, we showed that operation in the nonadiabatic regime, where there is mixed geometric- and dynamic-phase evolution, allows magnetic field sensitivity to be better than that of dynamic-phase magnetometry. We expect that geometric-phase AC field sensing will provide similar advantages to dynamic-phase magnetometry, although the experimental protocol (Berry sequence) will need to be adjusted to allow only accumulation of geometric phase due to the AC field. The generality of our geometric-phase technique should make it broadly applicable to precision measurements in many quantum systems, such as trapped ions, ultracold atoms, and other solid-state spins.

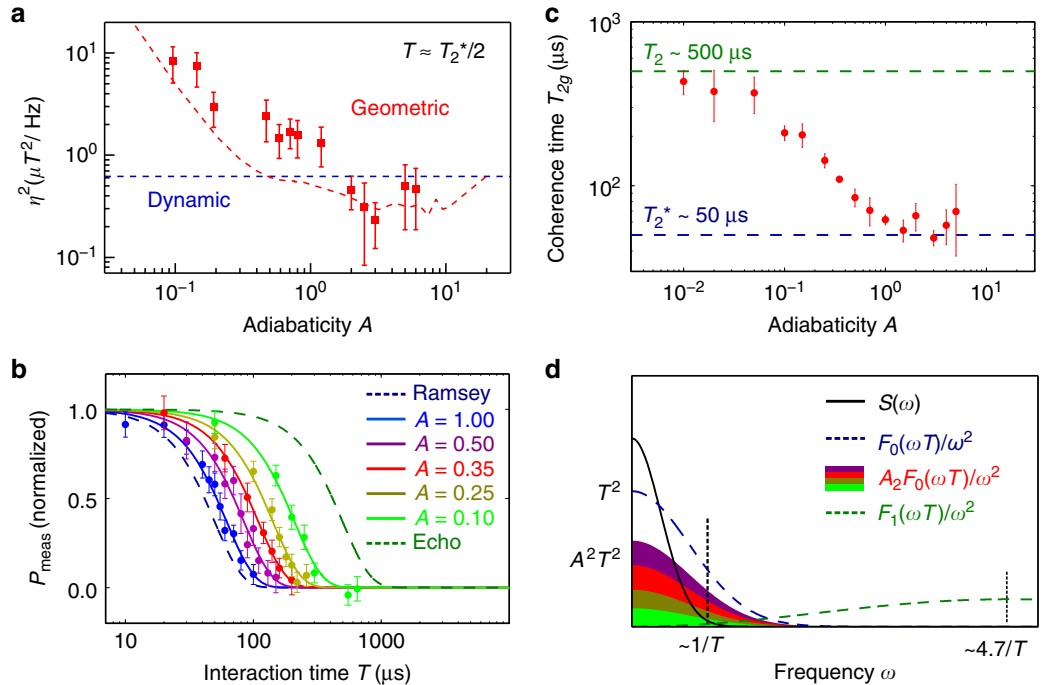

**Fig. 4** Improved geometric-phase coherence time and sensitivity in nonadiabatic regime. **a** Measured geometric-phase magnetic field sensitivity-squared (red squares) plotted against adiabaticity parameter $A$ using a fixed interaction time of $T \approx T_2^*/2$ at which the dynamic-phase Ramsey sequence gives optimal sensitivity (dashed blue line). Dashed red line shows geometric-phase sensitivity lower limit calculated by a numerical simulation assuming maximum signal contrast. The simulation does not include the contrast reduction due to hyperfine modulation. **b** Measured Berry sequence signal as a function of interaction time $T$ for various adiabaticity parameter values. Color dots are data; solid color lines are exponential fits to data $\sim\exp(T/T_{2g})^2$. Blue and green dashed lines indicate $T_2^*$ and $T_2$ decay of the dynamic-phase signal measured with a Ramsey and Hahn-echo sequence, respectively. **c** Measured geometric-phase coherence time $T_{2g}$ as a function of adiabaticity parameter $A$. Three regimes are observed: (i) For $A < 0.1$, $T_{2g} \sim T_2$, (ii) For $0.1 < A < 1.0$, $T_{2g} \sim T_2^*/A$, and (iii) For $A \sim 1.0$, $T_{2g} \sim T_2^*$. **d** Qualitative representation of contributions to the decoherence function (Eq. (3)) in the frequency domain: environmental noise spectral density function $S(\omega)$ (black line); dynamic-phase (spin-echo) filter function $F_1(\omega T)/\omega^2$ (dashed green line); and geometric-phase (Berry sequence) filter function $A^2 F_0(\omega T)/\omega^2$ (filled color area, same color-coding as in **b**), which vanishes in the limit $A \to 0$ and reaches the Ramsey sequence function $F_0(\omega T)/\omega^2$ (dashed blue line) in the limit $A \to 1$. Error bars represent one standard deviation of the results

## Methods

**NV diamond sample**. The diamond chip used in this experiment is an electronic-grade single-crystal cut along the [110] direction into a volume of $4 \times 4 \times 0.5$ mm$^3$ (Element 6 Corporation). A high-purity chemical vapor deposition layer with 99.99% $^{12}$C near the surface contains preferentially oriented NV centers. The estimated N and NV densities are $1 \times 10^{15}$ and $3 \times 10^{12}$ cm$^{-3}$, respectively. The ground state of an NV center consists of an electronic spin triplet with the $m_s = 0$ and $\pm 1$ Zeeman sublevels split by $2\pi \times 2.87$ GHz due to spin−spin interactions. Excitation with green (532 nm) laser light induces spin-preserving optical cycles between the electronic ground and excited states, entailing red fluorescence emission (637−800 nm). There is also a nonradiative decay channel from the $m_s = \pm 1$ excited states to the $m_s = 0$ ground state via metastable singlet states with a branching ratio of $\sim 30\%$. Thus, the amount of red fluorescence from the NV center is a marker for the $z$-component of the spin-state, and continuous laser excitation prepares the spin into the $m_s = 0$ state over a few microseconds. The spin qubit used in this work consists of the $m_s = +1$ and 0 ground states. Near-resonant microwave irradiation allows coherent manipulation of the ground spin states. The NV spin resonance lifetimes are $T_1 \sim 3$ ms, $T_2 \sim 500$ μs, and $T_2^* \sim 50$ μs.

**Confocal scanning laser microscope**. Geometric-phase magnetometry using single NV centers is conducted using a home-built confocal scanning laser microscope (Supplementary Fig. 1). A three-axis motorized stage (Micos GmbH) moves the diamond sample in three dimensions. An acousto-optic modulator (Isomet Corporation) operated at 80 MHz allows time-gating of a 400 mW, 532 nm diode-pumped solid-state laser (Changchun New Industries). An oil-immersion objective (×100, 1.3 NA, Nikon CFI Plan Fluor) focuses the green laser pulses onto an NV center. NV red fluorescence passes through the same objective, through a single-mode fiber cable with a mode-field-diameter of ~5 μm (Thorlabs), and then onto a silicon avalanche photodetector (Perkin Elmer SPCM-ARQH-12). The NV spin initialization and readout pulses are 3 μs and 0.5 μs, respectively. The change of fluorescence signal is calculated from $\Delta FL = FL^+ - FL^-$, where $FL^{\pm}$ are the fluorescence counts obtained after spin projection using a microwave π/2-pulse along the $\pm x$-axis, respectively. For each measurement, the fluorescence count $FL$

when the spin is in the $m_s = 0$ state is also measured as a reference. The temperature of the confocal scanning laser microscope is monitored by a 10k thermistor (Thorlabs) and stabilized to within 0.05 °C using a 15 W heater controlled with a PID algorithm.

**Hamiltonian parameter control system**. The Rabi frequency ($\Omega$) and phase ($\rho$) of the microwave drive field, as well as the applied magnetic field to be sensed ($B$), are key variables of this work. It is thus crucial to calibrate the microwave driving system and magnetic field control system beforehand. Microwave pulses for NV geometric phase magnetometry are generated by mixing a high frequency (~3 GHz) local oscillator signal and a low frequency (~50 MHz) arbitrary waveform signal using an IQ mixer (Supplementary Fig. 1). The Rabi frequency and microwave phase are controlled by the output voltage of an arbitrary waveform generator (Tektronix AWG5014C) (Supplementary Fig. 2). The microwave pulses are amplified (Mini-circuits ZHL-16W-43-S+) and sent through a gold coplanar waveguide (10 μm gap, 1 μm height) fabricated on a glass coverslip by photolithography. An external magnetic field for magnetometry demonstration is created by sending an electric current through a copper electromagnetic coil (4 mm diameter, 0.2 mm thick, $n = 40$ turns, $R = 0.25$ Ω) placed $h = 0.5$ mm above the diamond surface. The electric current is provided by a high-stability DC voltage controller (Agilent E3640A). To enable fine scan of the electric current with limited voltage resolution, another resistor with 150 Ω is added in series. Thus, a DC power supply voltage of 3 V approximately corresponds to $I = 0.02$ A, which creates an external field of $B = \mu_0 nI/4\pi h \sim 16$ G. One can determine the change of the external magnetic field as a function of DC power supply voltage $\Delta B(V)$ by measuring the shift of the resonance peak $\Delta f$ in the NV electron spin resonance spectrum using $\Delta f = \gamma \Delta B$. The result is $\Delta B/V = 0.50 \pm 0.01$ G V$^{-1}$ (Supplementary Fig. 3). Joule heating produced by the coil is $P = I^2 R \sim 10^{-4}$ W. The mass and heat capacity of the coil are about 0.15 g and 0.06 J K$^{-1}$, respectively. Thus, the temperature rise is at most 2 mK s$^{-1}$. Since the temperature coefficient of the fractional resistivity change for copper is 0.00386 K$^{-1}$[32], the change of resistance due to Joule heating is negligible.

**Numerical methods for geometric phase simulation.** All simulations of NV spin evolution in this work are carried out by computing the time-ordered time evolution operator at each time step.

$$U(t_i, t_f) = \hat{T}\left\{\exp\left(-i\int_{t_i}^{t_f} H(t)\,dt\right)\right\} = \prod_{j=1}^{N}\exp\left[-i\Delta t H(t_j)\right], \qquad (4)$$

where $t_i$ and $t_f$ are the initial and final time, respectively, $\hat{T}$ is the time-ordering operator, $\Delta t$ is the time step size of the simulation, $N=(t_f-t_i)/\Delta t$ is the number of time step, and $H(t)$ is the time-dependent Hamiltonian (Eq. (1)). In the simulation, we used $\Delta t = 1$ ns step size which is sufficiently small in the rotating frame. The algorithm is implemented with MATLAB®.

**Data and code availability.** The data and numerical simulation code that support the findings of this study are available from the corresponding author upon reasonable request.

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

## Acknowledgements

This material is based upon work supported by, or in part by, the U.S. Army Research Laboratory and the U.S. Army Research Office under contract/grant numbers W911NF1510548 and W911NF1110400. This work was performed in part at the Center for Nanoscale Systems (CNS), a member of the National Nanotechnology Coordinated Infra-structure Network (NNCI), which is supported by the National Science Foundation under NSF award no. 1541959. J.L. was supported by the ILJU Graduate Fellowship. We thank John Barry, Jeff Thompson, Nathalie de Leon, Kristiaan de Greve, and Shimon Kolkowitz for helpful discussions.

## Author contributions

K.A., C.B., and R.L.W. conceived and K.A. and J.L. designed the experiments. K.A., J.L., and H.Z. performed the experiments and processed the data. All authors analyzed the results. K.A., J.L., D.R.G. and R.L.W. wrote the manuscript.

## Additional information

**Competing interests:** The authors declare no competing interests.

