## [Peer Review File · Nature Communications]

Reviewers' comments:

Reviewer #1 (Remarks to the Author):

The article by Arai et al. introduces and demonstrates a modified geometric phase measurement protocol for magnetometry in particular with the electronic spin associated with the nitrogen vacancy center in diamond. Compared with a Ramsey sequence for low frequency magnetometry, the geometric phase measurement protocol has the advantage that the maximum field range and the sensitivity can be adjusted as independent parameters, enabling high sensitivity and high field range at the same time.

The article is clearly structured and very well written. The results are convincingly presented. I have the following questions and comments about the content which may require minor changes to the article.

- 1) The experiment was carried out with a NV defect hosted by a ^{12}C purified diamond layer yielding rather long T_2^* and T_2 times. How are the main conclusions, large field range and high sensitivity with geometric phase protocol relative to the Ramsey sequence, affected if the experiment was carried out with a defect in a diamond with natural abundance of ^{13}C and respectively shorter coherence times? In more general terms, are long T_2^* times generally required to have an advantage of the geometric phase over the dynamic protocol?
- 2) Could the authors comment on the measurement bandwidth of the present geometric phase approach and compare it to the Ramsey sequence? How does the bandwidth depend on the adiabaticity parameter?
- 3) Can the authors comment on the possible extension of the geometric phase approach to sensing of ac fields? May similar improvements to the CPMG and XY-N protocols be expected?

Minor comments:

- 4) In line 84, I find the notation of $B=0$ and the subsequent assumption that only two of the NVs ground state spins are considered somewhat confusing. This consideration may be clarified further.
- 5) In the equation for B_m in line 68 seems to be a typo, which may have occurred while compiling the document.
- 6) In line 475, the reference is to fig. 4a, which however should be 4b I assume.

I'm convinced the geometric phase protocol will have a significant impact in the community and hence I recommend publication in Nat. Comm. after addressing the above comments and questions.

Reviewer #2 (Remarks to the Author):

This paper reports geometric magnetometry based on a single spin. Such magnetometry, harnessing geometric phases, can reach large measurement range as well as high sensitivity. The authors also explore the property of the geometric magnetometry as a function of the adiabaticity. The experiment performed on the electron spin of a nitrogen-vacancy center in diamond demonstrates the principal features of the geometric magnetometry.

Overall, this work shows that geometric phases can be applied for quantum sensing, and the paper would be suitable for publication in Nature Communications. The authors should address the following issues before the publications.

1. The geometric phase is induced by effective signal "B" with off-resonance driving. The direction of effective signal B is alternated before and after a π pulse at the center of a Berry sequence.

Also, the detection time is strictly controlled to whole periods N . So, it should be discussed how to overcome these limitations and apply this method to a realistic DC or AC magnetic signal.

2. In a situation, B_{\max} and η is proportional to $\omega^{0.9} N^{0.52} T^{0.02}$ and $\omega^{1.2} N^{-0.92} T^{0.46}$. Is it a general case and is there a physical explanations for these numbers?

3. A work on Appl. Phys. Lett. 112, 252406, 2018, which is titled with Robust high-dynamic-range vector magnetometry with nitrogen-vacancy centers in diamond, provides an easier methods to realize high-dynamic range without effect the sensitivity. It should be cited and please address the advantages of this method compared to the previous one.

4. The format errors should be corrected in the manuscript, such as the title of citations should be deleted [citations 28, 29]

Reviewer #1 (Remarks to the Author):

The article by Arai et al. introduces and demonstrates a modified geometric phase measurement protocol for magnetometry in particular with the electronic spin associated with the nitrogen vacancy center in diamond. Compared with a Ramsey sequence for low frequency magnetometry, the geometric phase measurement protocol has the advantage that the maximum field range and the sensitivity can be adjusted as independent parameters, enabling high sensitivity and high field range at the same time.

The article is clearly structured and very well written. The results are convincingly presented. I have the following questions and comments about the content which may require minor changes to the article.

1) The experiment was carried out with a NV defect hosted by a ^{12}C purified diamond layer yielding rather long T_2^* and T_2 times. How are the main conclusions, large field range and high sensitivity with geometric phase protocol relative to the Ramsey sequence, affected if the experiment was carried out with a defect in a diamond with natural abundance of ^{13}C and respectively shorter coherence times? In more general terms, are long T_2^* times generally required to have an advantage of the geometric phase over the dynamic protocol?

Response: In general, long T_2^* time is not required for the geometric phase technique to have an advantage over the dynamic phase protocol. For any given T_2^* time, the geometric phase protocol can have similar or better sensitivity, as well as much larger field range, compared to the dynamic phase protocol. This superior performance is because the geometric phase protocol has 3 control parameters (Ω , N , T), which can be used to manipulate sensitivity and field range independently; in contrast, the dynamic phase protocol has only one control parameter (T). Even when T_2^* is short, one can optimize the other control parameters (Ω and N) to maintain good sensitivity with the geometric phase protocol; while the improved field range will not be affected since it only depends on N and Ω . These points are discussed in the revised manuscript main text and supplement.

2) Could the authors comment on the measurement bandwidth of the present geometric phase approach and compare it to the Ramsey sequence? How does the bandwidth depend on the adiabaticity parameter?

Response: The measurement bandwidth for a Ramsey sequence scales as $\sim 1/T_2^*$ (in our experiment ~ 20 kHz). The geometric phase coherence time depends on the adiabaticity parameter A . Thus, for $A < 0.1$ or $A > 1$, the measurement bandwidth scales as $\sim 1/T_2$ (in our experiment ~ 2 kHz for the Hahn-echo T_2); and for $0.1 < A < 1$, the measurement bandwidth scales as $\sim A/T_2^*$. This behavior can be understood via spectral density theory, as discussed in the revised manuscript main text and supplement.

3) Can the authors comment on the possible extension of the geometric phase approach to sensing of ac fields? May similar improvements to the CPMG and XY-N protocols be expected?

Response: In principle, geometric phase AC field sensing is possible. However, one cannot naively apply the present geometric phase approach to CPMG or XY-N protocols. Instead, one must (i) embed the geometric phase protocol in a Ramsey sequence in order to cancel out the accumulation of dynamic phase due to the AC field, and (ii) alternate the direction of phase rotation of the geometric phase pulse. We expect that, as with geometric DC field sensing, geometric phase AC field sensing will provide a larger field range than a dynamic phase protocol. We edited the outlook section of the main text to provide a brief discussion of the potential for geometric phase AC field sensing.

Minor comments:

4) In line 84, I find the notation of $B=0$ and the subsequent assumption that only two of the NVs ground state spins are considered somewhat confusing. This consideration may be clarified further.

Response: We revised the text to clarify that in our experiments we use both (i) a constant bias magnetic field (≈ 9.6 mT) to lift the degeneracy of the NV spin transitions; and (ii) a small signal magnetic field B of typical magnitude ~ 100 microtesla.

5) In the equation for B_m in line 68 seems to be a typo, which may have occurred while compiling the document.

Response: We corrected this typo.

6) In line 475, the reference is to fig. 4a, which however should be 4b I assume.

Response: We corrected this typo.

I'm convinced the geometric phase protocol will have a significant impact in the community and hence I recommend publication in Nat. Comm. after addressing the above comments and questions.

Reviewer #2 (Remarks to the Author):

This paper reports geometric magnetometry based on a single spin. Such magnetometry, harnessing geometric phases, can reach large measurement range as well as high sensitivity. The authors also explore the property of the geometric magnetometry as a function of the adiabaticity. The experiment performed on the electron spin of a nitrogen-vacancy center in diamond demonstrates the principal features of the geometric magnetometry.

Overall, this work shows that geometric phases can be applied for quantum sensing, and the paper would be suitable for publication in Nature Communications. The authors should address the following issues before the publications.

1. The geometric phase is induced by effective signal “B” with off-resonance driving. The direction of effective signal B is alternated before and after a π pulse at the center of a Berry sequence. Also, the detection time is strictly controlled to whole periods N. So, it should be discussed how to overcome these limitations and apply this method to a realistic DC or AC magnetic signal.

Response: The geometric phase DC magnetometry protocol does not alternate the direction of the effective signal B before and after a π pulse; nor is the detection time limited to whole periods N. Rather, as illustrated in Supplementary Figure 2, the effective signal B is constant over the whole measurement; whereas phase rotation of the NV Larmor vector $R(t)$ is alternated in order to accumulate geometric phase and suppress dynamic phase. We revised the text to make this point more clearly, and also provide a brief discussion of the potential for geometric phase AC field sensing.

2. In a situation, B_{\max} and η is proportional to $\Omega^{0.9} N^{0.52} T^{0.02}$ and $\Omega^{1.2} N^{-0.92} T^{0.46}$. Is it a general case and is there a physical explanations for these numbers?

Response: The experimental results for the scaling of B_{\max} and η with the control parameters (T, N, and Ω) is consistent, within experimental uncertainty, with analytical expressions derived for the adiabatic limit, as found in the supplementary material. Intuitively, B_{\max} can be understood as the furthest minimum of the chirped geometric phase magnetometry curve, and sensitivity can be calculated at the maximum slope ($B \sim 0$) of the magnetometry curve. These points are made in the revised text.

3. A work on Appl. Phys. Lett. 112, 252406, 2018, which is titled with Robust high-dynamic-range vector magnetometry with nitrogen-vacancy centers in diamond, provides an easier methods to realize high-dynamic range without effect the sensitivity. It should be cited and please address the advantages of this method compared to the previous one.

Response: We added a reference to this work, and noted that this closed-loop continuous measurement technique can be expected to have poorer sensitivity relative to our pulsed geometric phase approach.

4. The format errors should be corrected in the manuscript, such as the title of citations should be deleted [citations 28, 29]

Response: We corrected these errors.

REVIEWERS' COMMENTS:

Reviewer #1 (Remarks to the Author):

All my previous comments were addressed in an appropriate manner, and hence I recommend the article for publication in its present form.

Reviewer #2 (Remarks to the Author):

The authors have properly addressed the points in my first report. I recommend for publication in Nature Communications.